# Aflatoxin Binders in Foods for Human Consumption—Can This be Promoted Safely and Ethically?

**DOI:** 10.3390/toxins11070410

**Published:** 2019-07-14

**Authors:** Sara Ahlberg, Delia Randolph, Sheila Okoth, Johanna Lindahl

**Affiliations:** 1Department of Biosciences, International Livestock Research Institute, P.O. Box 30709, Nairobi 00100, Kenya; 2Department of Food and Environmental Sciences, University of Helsinki, P.O. Box 66, FI-00014 Helsinki, Finland; 3School of Biological Sciences, University of Nairobi, P. O. Box 30197, Nairobi 00100, Kenya; 4International Livestock Research Institute, 298 Kim Ma Street, Ba Dinh District, Hanoi, Vietnam; 5Department of Medical Biochemistry and Microbiology, Uppsala University, P.O. Box 582, 75123 Uppsala, Sweden; 6Department of Clinical Sciences, Swedish University of Agricultural Sciences, P.O. Box 7054, 75007 Uppsala, Sweden

**Keywords:** Aflatoxins, binding, food safety, biocontrol, food discipline

## Abstract

Aflatoxins continue to be a food safety problem globally, especially in developing regions. A significant amount of effort and resources have been invested in an attempt to control aflatoxins. However, these efforts have not substantially decreased the prevalence nor the dietary exposure to aflatoxins in developing countries. One approach to aflatoxin control is the use of binding agents in foods, and lactic acid bacteria (LAB) have been studied extensively for this purpose. However, when assessing the results comprehensively and reviewing the practicality and ethics of use, risks are evident, and concerns arise. In conclusion, our review suggests that there are too many issues with using LAB for aflatoxin binding for it to be safely promoted. Arguably, using binders in human food might even worsen food safety in the longer term.

## 1. Aflatoxins in Developing Country Food Chains with a Special Focus on Kenya

Mycotoxins, including the important fumonisins, trichothecene toxins, zearalenone, and especially aflatoxins, have caused great concern in African and especially Kenyan markets over the last four decades. These mycotoxins are widespread, contaminating cereals, potatoes, bananas, cotton, and other plants. Additional mycotoxins, such as ochratoxins and patulin, are found in coffee, apples, and citrus fruits [1].

Aflatoxins are an important group of mycotoxins because there is strong evidence of their severe health impacts, causing liver cancer, especially among hepatitis B–positive people [2,3,4]. Extended exposure is implicated in immunodeficiency, immunosuppression, stunting, kwashiorkor, and interference with the metabolism of micronutrients in children [4]. High prevalence of aflatoxins in staples and consequently chronic exposure is common in regions where control and monitoring systems are poor and regulations are not enforced. Many studies find aflatoxins are present in high levels in both feed and food chains in Africa, exposing consumers to aflatoxins, especially through staple foods [5].

Aflatoxins are produced by toxin-producing fungi *Aspergillus*, but fungal growth does not necessarily entail toxin production. Naturally occurring, there are non-toxic and toxic strains that produce aflatoxins at different levels [6]. Fungal growth and aflatoxin production are driven by climatic conditions. Any pre-harvest contamination of maize with *Aspergillus* fungi can lead to the accumulation of considerable aflatoxin levels when post-harvest conditions are adverse. However, post-harvest preventive measures against fungal contamination are more common than pre-harvest measures [7]. 

Acute aflatoxicosis is caused by consumption of large amounts of aflatoxins. This has occurred repeatedly in Kenya and other countries resulting in outbreaks with hundreds of human and thousands of animal deaths in the worst cases [8,9,10]. These widely reported cases have led to increased public concern and stimulated research efforts, policy changes, and investments into the research of suitable and effective mitigation interventions, and increased awareness of safety measures. However, these efforts have not been shown to decrease either the prevalence nor the dietary exposure to aflatoxins [9].

Kenya, a hot-spot of aflatoxins, has frequent, high, and not consistently improving prevalence of aflatoxins in staples and animal feeds. Aflatoxin studies report high proportions of cereals and feeds contaminated to some extent, and many samples exceed the allowable limits [8,11,12,13,14]. Likewise, fumonisins are found in almost all crops, often in co-occurrence with aflatoxins [8,15,16,17,18,19,20,21,22]. In consequence of the crop and feed contamination, almost all cattle milk is contaminated with aflatoxins [8,11,12,13,22,23,24].

Compared with other common foodborne hazards, aflatoxins are unusual because they can be formed only as a result of fungal infestation, usually at the farm level. This is exacerbated and spread by poor storage conditions. Once the aflatoxins are introduced, products are contaminated, and, if not removed from the chain at the control point when detected, they move further along the food chain and through processing. Heat treatments used in food production cannot eliminate the formed aflatoxins. Aflatoxins and other mycotoxins are invisible and can be detected only with modern analytical methods. However, if visible *Aspergillus* mould is present, this is an indicator of risk. The lack of control and monitoring in developing regions enables the supply of contaminated crops to reach the consumers. 

Exposure to aflatoxins can be assessed through blood samples detecting albumin adducts or through detection of metabolites in milk or urine. Surveys report a wide range of exposure levels, from nondetectable to very high. Aflatoxin levels reported from Kenya during the 2010 outbreak were the highest ever reported (even up to 1200 pm/mg albumin) [5,10,25,26,27]. An indirect assessment of human exposure is the contamination level in food products. 

Poverty is associated with poor availability and quality of foods, and this is also associated with aflatoxin exposure levels. Higher aflatoxin exposure levels were associated with the lowest socio-economic conditions in a study in Kenya, although all the women sampled were exposed [28]. In Africa, many small-holder farmers are women, who farm mostly for household consumption and informal markets and lack resources to avoid aflatoxin exposure.

Many mitigation methods have been suggested, from farm- to consumer-level interventions. Wild and Gong [29] have listed reasons for failures in aflatoxin control strategies. This list, which is relevant still a decade later, includes
The perceived value of interventions may be low and a main reason for this could be the broken food chains where farmers, producers, and supply chain actors are working in isolation from each other, their efforts are not clearly rewarded, and (probably even more importantly) negligence is not sanctioned;Toxins are invisible and tasteless, making them difficult for both producers and consumers to assess;Control is required along the food chain in several points, and currently, the ability to cover the food chains throughout by food inspectors is poor in developing regions;The highest exposure may be in informal markets where regulations and control do not reach;Aflatoxins are a multidisciplinary problem of agriculture, public health, and economics.

Staple foods in Africa are the most contaminated with aflatoxins and other mycotoxins. Promotion of healthy diets and diversification of food sources in the diet, e.g., increased diversity of legumes and vegetables, could be one significant way to decrease the levels of exposure. However, most people in Africa cannot afford diverse diets. Nonetheless, diversification of nutrient sources should be promoted, not only from the contamination exposure point of view, but also from agricultural and environmental diversity and nutritional perspective. Focus on staples and fungus-resistant maize can further decrease the promotion of diversity in diets and in agriculture, promoting further monocropping leading to decreased biodiversity levels, which are declining globally in alarming levels. 

## 2. Binding of Aflatoxins as a Biocontrol Method

Novel approaches and new intervention methods focusing strongly on finding solutions to aflatoxin contamination have been called for. Risk mitigation and food safety improving measures have attracted funding resources, leaving other issues and problems, including other mycotoxins, behind. For example, aflatoxin research has benefited from a level of donor support disproportionate to the health burden it causes. According to the World Health Organisation and World Bank, aflatoxins are a relatively minor contributor to the overall health burden of foodborne disease [30,31], but the WHO report only includes the burden from hepatocellular carcinomas.

A specific approach to aflatoxin control is the use of aflatoxin-binding agents in foods. The principle is as follows: aflatoxins, which have contaminated foods, can be bound to an agent to mitigate the aflatoxin-induced health risks after consumption. Binders include bacteria cells, yeasts, proteins, and clays; the latter have been especially analysed for use in animal feeds. The hypothesis is that the binding agent and the bound toxin would pass through the gastrointestinal tract without, or at least with less, uptake and thus less damage caused by the toxins. Binding with lactic acid bacteria (LAB) is discussed below. Some other organic binding agents analysed have been yeasts [32,33].

Evidence of the binding ability of aflatoxins with LAB cells has been shown through a number of studies in laboratory conditions, some with 100% binding efficiency [32,34,35]. Binding is speculated to be an instant phenomenon [32,36,37,38,39], but also binding levels have been observed to increase over time [32,36,37,39,40,41,42,43].

Contrary to observed instant binding, some studies have reported no immediate binding at all [43,44]. Govaris [45] also noted several contradictions among the studies since the 1980s. Conflicting results have also been reported from storage studies. While Ahlberg [46] observed both increased binding over time and release of aflatoxins back to the matrix during 21-day trial, Barukčić [47] and Govaris [45] reported binding levels to remain the same even for 21 days. Sokoutifar [44] recorded large amounts of aflatoxins bound to LAB strains up to 30 days at 21 and 37 °C. In practice, however, such high temperature conditions cannot be attained due to food integrity and safety risks.

While some authors reported increasing binding efficiency of LAB with increased aflatoxin concentration, others have reported decreasing effects or no difference or even both [34,41,45,46]. Binding has been shown to be dependent also on the concentration of the LAB cells [34,35,48].

Viability of bacteria strains has been considered a significant factor in binding. However, both viable and non-viable LAB strains have performed better in binding over the other in different studies, and no difference between the two has also been found [36,37,42,48,49,50]. These results have not brought clarity to the binding mechanisms, whether the binding effect is due to physical binding or influenced by the components produced by the bacteria.

Other factors affecting binding efficiency have been reported to occur in different food matrices such as milk or yoghurt, possibly explained by the compounds in the matrix [40,45], lower pH [45], or even higher pH [37,48]. In conclusion, external conditions seem to strongly affect the binding ability of aflatoxins by LAB.

One factor to consider in the binding analysis is the stability of the bound complex. Even simple washing can release 20–70% of the initially categorized bound aflatoxins back to detectable forms [32,34,36,37,39,51,52]. The stability of the formed bond is an important factor to assess the suitability of binding agents in food systems to reduce the harmful effects of aflatoxins. 

As LAB are commonly used in dairy fermentation, the binding efficiencies of milk components have been studied. The milk protein casein is often speculated to be a binding agent in milk, the cheese making process being an indicator for this phenomenon due to the separation of whey and casein fractions. Aflatoxin binding has been concluded both to increase and decrease during cheese making [53]. In some of the binding studies, the controls without LAB cells show very low binding and reduction in aflatoxin shares (2–5%) compared with the binders [43,44]. These findings do not support the binding of milk components or casein to aflatoxins to be anything significant.

One of the first studies in binding concluded LAB removed as much as 80% of aflatoxins during cooking [17], which probably resulted a flourishing of interest in this research sector. Scientific evidence shows good potential in binding methods if certain criteria for evaluation are selected. However, when considering binding from wider perspective, serious concerns and problems arise, which have not been discussed or critically reviewed within these applications.

## 3. Challenges with Interpreting the Results of Binding Aflatoxins with LAB

Binding mechanisms and efficiency factors for LAB are not clearly understood and are considered still speculative in publications on binding. There seems to be no predictable factor affecting the binding efficiency and stability, resulting in the unpredictability and uncontrollability of the binding process. Optimal conditions for controlled and predictable binding have not been found. One factor can enhance binding shares in one study, but the same factor decreases the binding shares in another study. For example, the level of aflatoxin concentration is speculated to be one major factor in binding efficiency. It is especially important to bear this in mind because, as aflatoxins are contaminants, the levels and prevalence are unpredictable and vary significantly between batches, commodities, regions, and seasons. The approach to increase the safety of foods with aflatoxin binding with LAB cannot depend on the uncontrollable contamination level.

The binding analyses follow fairly simple procedures. Binders and LAB are mixed and possibly incubated in a liquid media (milk, broth, PBS, etc.) with aflatoxins. The mixture is then centrifuged, and the pellet is considered containing the bound aflatoxins attached to the LAB, as the free, unbound aflatoxins are considered remaining in the supernatant, the liquid media. It is possible that in this method the aflatoxins can be “trapped”: physically pulled down by the other components of the binding analysis matrix to the pellet during centrifugation. This is even more likely when fermentation is taking place: LAB produce exopolysaccharides, high in molecular weight and large in structure constructing extracellular polymeric substances (EPSs) with proteins. These are partly responsible for the thickening of the product during fermentation. As any high molecular component will be pelletized during centrifugation, so are the fermenting products, which then can easily trap the aflatoxins and further falsely be detected as “bound”. 

For food safety purposes, both the binding efficiency and the stability of the formed bond are relevant. A weak formed bond releasing the aflatoxin would not have mitigation potential, despite the initial binding efficiency. If the binding phenomenon is only temporary, the suitability as a food safety method will not be relevant due to the uncontrollable conditions and risks induced. Several studies have reported how different levels of aflatoxins are released from formed aflatoxin and LAB complex under different conditions [32,34,36,37,39,51,52].

One major flaw in aflatoxin binding studies is the over-optimistic rhetoric used in the studies and conclusions. A number of studies observed binding in laboratory conditions with limited replications yet concluded it to be a suitable method of improving food safety. These conclusions contradict standard approaches to food safety measures, guidance, and regulations development, which would not support use of additives on the basis of inconclusive evidence. The phrase “aflatoxins could be removed” is often used in aflatoxin-binding studies, but in practice, the aflatoxins are still present in the food at the original levels, whether bound or not.

The analysis of binding of aflatoxins by LAB raises a question about the suitability of the methods. Aflatoxin contamination methods for screening contamination levels from foods uses the same analyses as the binding methods. These results of aflatoxin screening in different studies can sometimes show even higher aflatoxin contamination levels for the fermented food and milk products, which are incompatible with bound aflatoxins [23]. To further speculate, in principle, if the binding of aflatoxins to LAB, to milk components, or other food components occurs, all the analysed levels of aflatoxins from foods would be higher in reality than the given results indicate. Alternatively, it could be implied that the analysis methods for food contamination levels are not appropriate for the binding trials.

## 4. Big Picture—Safe Food for All

Promotion of aflatoxin binding at the consumer level of the food chain can signal to the producers and operators that the production of unsafe foods is acceptable as the problem could be solved later on. Such new principles can be extremely difficult to reverse later on, especially in poorly regulated markets. The awareness, knowledge, and practice of safety measures about mycotoxins and aflatoxins among farmers [7], producers, and consumers is limited [12,24,54], and promoting a method with uncertainties could easily create new misunderstandings and misperceptions of the causalities behind the contamination patterns and induced health risks. 

Development of binders has taken a highly technological approach with little consideration to ethical, political, consumer acceptability, or legal implications. Yet using binders raises serious concerns and questions about risk, trade-offs, and entitlements that have not been discussed, let alone addressed. Without thoroughly understanding these aspects, it is likely that even if LAB is found to be technologically effective, it may not be adopted, or that if adopted, it could have unintended negative consequences.

Today, poor consumers patronising informal market chains cannot enjoy the same fundamental right to safe food as the wealthy consumers in formal markets in high income countries. In developing countries, market regulations, although inadequately implemented, mainly cover the formal markets, leaving informal markets unregulated [23,55]. In Kenya, among branded products sold in formal markets, lowest priced maize was 25% less likely to meet regulatory requirements for aflatoxins than the highest priced products [56]. Some indicators show that the situation might be worse in informal markets, but no systematic comparison has been done between the maize products sold in formal and informal markets in Kenya [56]. Aflatoxin exposure from milk among low-income consumers in urban Nairobi is higher than among mid-income consumers due to the higher aflatoxin levels in products sold in low-income areas and the higher milk consumption [23].

One effect of promoting fermentation with LAB to reduce aflatoxins in the informal sector could be the development of double standards in the food safety and food production systems. In principle, promoting different standards and procedures in different markets will create problems later in the upgrading and formalisation of traditional markets. 

While aflatoxins are present in large parts of the world, high exposure levels in humans are mainly a problem in developing regions, and worst among poor purchasers. These people have often less access to information, and their understanding about the options, alternatives, and the relationship between actions at the beginning of the production chain and the consumption level may be lacking. Consumers in informal markets have limited access to the regulated markets without full market structure change. 

Promotion of the use of aflatoxin binders in foods could potentially create new layers of problems. These have not received attention because the solution has been developed from a perspective of scientific functionality. Aflatoxins are by far the most studied mycotoxins [57], and when other mycotoxins start to gain more publicity, aflatoxin binding may appear inadequate as a solution. The role of social sciences should be promoted to create collaboration and multidisciplinary academic knowledge to develop new and suitable ways to work against aflatoxins and increase food safety [9].

Notably aflatoxin binding research has approached the issue from a one-component “silver bullet” solution instead of focusing on comprehensive food safety solutions at the farm and value chain level mitigating all the mycotoxins. Other mycotoxins are prevalent and occur together with aflatoxins. The binding solution is a rather simplified solution for a complex problem formed due to several factors and enchased by insufficient practices.

## 5. Ethical Assessment to Improve Food Safety with Binders in Human Foods

“Humans have a right to food free from mycotoxins that could cause significant health risk”.

Declaration by the United Nations Environment Programme (UNEP) and the World Health Organization (WHO) International Programme on Chemical Safety (IPCS) [58]

An aflatoxin binding approach to foods inevitably requires testing and efficient analytical methods at consumer level. This is challenging for a number of reasons. First of all, promoting aflatoxin binding at the consumer level assumes people will deliberately be exposed to aflatoxins from contaminated foods and food products to assess the effectiveness, accepting something many citizens feel to be unacceptable. Second, aflatoxins are more of a problem in poorly regulated countries than in developed regions, especially among the poorest consumers.

Aflatoxins are one of the most regulated contaminants with allowable legal limits in commodities [59,60]. The role of the European Union in trading has pushed EU limits to be followed and adopted in regions with limited resources, creating a situation where limits are strict but resources are scarce to implement, monitor, and control the set limits. Also, the Codex Alimentarius has recommended limits for mycotoxins and aflatoxins, which can be adopted to national legislation [61]. The limits, whether reasonable and realistic from economic and trade perspective, are set to harmonize the safe food production systems to ensure the safety of the products.

The application of binders in human foods is in conflict with the principle of developed food safety regulations, set allowable limits, and regulation implementation and compliance by the operators. Even an emergency application can add new problems to the fragile and developing food safety systems. Officially approving a binder application in foods would be politically ambiguous. It is highly unlikely that developed regions, with strong regulatory systems, would allow aflatoxin binders at the consumer level as it is strongly against the principles of the current food regulatory systems. Implementing such binders legally only in developing regions with poor regulatory systems would raise concerns in terms of promoted double standards.

Clay supplements and LAB have been tested in human trials aimed to be used during an emergency aflatoxins outbreak situations [62,63]. In already poorly regulated regions, it would be challenging to keep the promotion of daily food safety measures and good practices separated from promoting a temporary solution or a quick-fix. It raises the concern at what threshold level would such an emergency outbreak be announced for the binding application in human foods to be “legal” or allowed. For example, in Kenya, many foods continuously contain aflatoxins above the allowable limits. Would high aflatoxin prevalence above legal limits permit the usage of binders in a specific time and region? Instead of supplementing binders to people in an aflatoxin emergency situation to encourage them to eat the contaminated, potentially high-risk maize, it would be more ethical to provide the replacement of safe maize for consumption.

## 6. Who Can Choose What to Eat?

A food safety method to control harmful contaminants should be robust, reliable, and functional in all conditions the contaminants are present. Also, to be suitable for the purpose, the methods needs to be available, feasible, understandable, and acceptable to the end users.

As the European Union is the largest economy in the world and a major trading partner for many countries, the EU legislation and standards are relevant globally. The EU has strict standards to ensure food safety and comprehensive regulation of practices to ensure the safety and quality of the products. It should be highlighted that set standards and limits alone cannot create food safety, but the comprehensive food industry system from farm practices, through processing to consumers, all controlled and monitored by relevant institutions, can create a chain of controlled and traceable practices. This element of comprehensive approach is lacking in poorly regulated and in informal markets.

Where the legislation is set and executed throughout the practices in food production to protect consumers, unsafe products rarely enter the market chains and can be recalled if necessary. The binding application idea is also related to the food security status, but can the science and research community promote it in regions where people who have no institutional food safety protection, allowing them to consume foods that in regulated regions would be categorised unsafe and not fit for consumption? These are fundamental issues that should be discussed before any binding applications are taken to further testing. Figure 1 illustrates the separation between informal and formal markets and the most likely binder application channel and consequences.

Consumers and end-users have very little influence on aflatoxin levels. Would consumers accept contaminated foods and milk for consumption with binding methods compared with better management at the farm- and supply chain–level to prevent the contamination altogether? Would poor and less-informed consumers be more approving toward the binding methods than informed, knowledgeable consumers who have more resources to understand the production chains and the consequences of the practices?

Judging from past trends, it is unlikely that the food safety standards and measures will be lightened. Consumers are increasingly conscious, information is ever more readily available, and consumers are demanding safer, high-quality foods produced sustainably, ethically, and fairly. Enabling and promoting the development of different food standards and measures in informal market sectors or poorly regulated regions is a very questionable approach to food safety, and the acceptability of binding applications should be brought to wider discussions from laboratories and the research community. 

One of the most important questions in the binding applications should be, would you take it? 

## 7. Suggestions for Way Forward

Using LAB to bind aflatoxins in foods may pose greater short- and long-term risks than benefits. Most important aspects are related to regulations, acceptability, and the creation of double standards when harmonized systems and merged markets are needed. Use of binding agents in foods contradicts all the existing principles and regulations set to ensure food safety. If such a method is promoted, the efforts to combat the aflatoxin problem at farm level and throughout the value chain, to eliminate and reduce the contaminants, could be compromised.

Aflatoxin control is not simple and needs a comprehensive approach covering food safety and economic development to address overall good farming and food production practices. Currently, food safety promotion through binders is discussed as an isolated factor, a magic bullet, to solve the problem. Over-reliance on technological solutions and inadequate attention to legal, ethical, political, and behavioural aspects of technologies as well as unintended consequences reduces the likelihood that agricultural innovations will have beneficial health and development outcomes. Now is the time to start addressing these neglected and important aspects of aflatoxin control.

Aflatoxin problems are prevalent especially in staples, and promoting diverse diets could reduce the exposure, especially from maize. Basically, all measures come with a cost, but creating new systems to promote increasing diversity in diets would directly contribute to diversity in crops in farming, creating resilience against climate change and unpredictable conditions. Promoting new value chains for staples and for a larger variety of plant and animal source foods can create new income sources for farmers while contributing to improved diets and decreased aflatoxin intake, directly contributing to a decreased public health burden from unsafe foods and unhealthy diets. When people become richer, they naturally diversify their diets, and aflatoxin exposure reduces. So, the promotion of development through economic and agricultural policy may be an indirect way of ending the scourge of aflatoxin [64]. Other public health approaches such as hepatitis B vaccination also have potential. Finally, the authorities’ role to ensure the food safety in poorly regulated regions covering both informal and formal markets, but also promoting the merge of the two, should be strengthened significantly.

In final conclusion, there are too many issues with the aflatoxin binding methodology and results for it to be promoted. This review also highlights that binders for humans may be counter-productive for food safety.

## Figures and Tables

**Figure 1 toxins-11-00410-f001:**
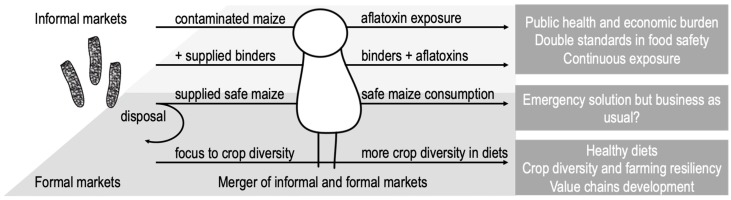
The most likely application chain for binder method applied in informal markets focusing strongly to the consumer actually taking the risk. Implementation of binding method in formal markets would be highly unlikely as the approach conflicts strongly against the regulatory allowable limits set to the aflatoxins. Informal and formal markets currently are not equal and should be merged into formal markets to enable the same food safety standards, economic growth, and new value chains in one coherent food production system.

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
