# Peer review of "Aflatoxin Binders in Foods for Human Consumption—Can This be Promoted Safely and Ethically?"

_toxins, 2019, doi:10.3390/toxins11070410_

Reviewer 1 Report

The authors provide a critical opinion on the use of aflatoxin binders in food and feed as it may be counter-productive for food safety.

Lines 34-36 : "Acute aflatoxicosis is caused by consumption of large amounts of aflatoxins. This has occurred several times in Kenya and other countries over the years resulting in outbreaks with hundreds of 35 deaths in the worst cases [7–9]".

Regarding the effects of aflatoxins on human and animal health the authors should mention a very thorough review written by Wu, Groopman and Pestka (2014) on the impacts of mycotoxins, including aflatoxins, on human health. Chronic consumption of food contaminated with aflatoxins causes growth stunting in children, liver and lung cancer, and impairment of immune system. Wu F, Groopman JD, Pestka JJ. Public health impacts of foodborne mycotoxins. Annu Rev Food Sci Technol. 2014;5:351-72.

Lines 46-51

Describing aflatoxin qualities it should be mentioned that these mycotoxins are heat stable and usually are not being destroyed during processing.

Are the biocontrol strategies such as "Aflaguard" being applied in Kenya? This could be an efficient measure to reduce aflatoxins in maize and peanuts. Aflaguard application does not always reduce the amount to zero, however the levels typically lower on average than in the untreated crop. 

Lines 164 - 174: the description of a general method for aflatoxin binding studies should be placed in the beginning of a chapter not by the end.

Lines 198-199: Could you provide data on poor vs wealthy consumer have different exposure to aflatoxins through food? Do you have some numbers to provide a clear picture? What are the differences in aflatoxin contaminated foods between informal and formal markets? Are there data of the surveys for these two markets?

What are the side effects of using aflatoxin binders in food and feed, especially for clay supplements and LAB?

Lines 233-240: Indicate what are the limits of aflatoxins in food in Europe, North America, and Africa.

Author Response

The authors provide a critical opinion on the use of aflatoxin binders in food and feed as it may be counter-productive for food safety.

Lines 34-36 : "Acute aflatoxicosis is caused by consumption of large amounts of aflatoxins. This has occurred several times in Kenya and other countries over the years resulting in outbreaks with hundreds of 35 deaths in the worst cases [7–9]".

Regarding the effects of aflatoxins on human and animal health the authors should mention a very thorough review written by Wu, Groopman and Pestka (2014) on the impacts of mycotoxins, including aflatoxins, on human health. Chronic consumption of food contaminated with aflatoxins causes growth stunting in children, liver and lung cancer, and impairment of immune system. Wu F, Groopman JD, Pestka JJ. Public health impacts of foodborne mycotoxins. Annu Rev Food Sci Technol. 2014;5:351-72.

Reference added to lines 21-> where health impacts are discussed: Aflatoxins are an important group of mycotoxins because there is strong evidence of their severe health impacts, causing liver cancer especially among hepatis B positive people [2–4].

Lines 46-51

Describing aflatoxin qualities it should be mentioned that these mycotoxins are heat stable and usually are not being destroyed during processing.

 Added for clarity: Heat treatments used in food production can’t eliminate the formed aflatoxins. (Reviews report inconsistent results with some studies indicating household processing involving heat can reduce, but not eliminate aflatoxins).

Are the biocontrol strategies such as "Aflaguard" being applied in Kenya? This could be an efficient measure to reduce aflatoxins in maize and peanuts. Aflaguard application does not always reduce the amount to zero, however the levels typically lower on average than in the untreated crop. 

Our manuscript initially covered feeds and aflatoxin control in farm/production level. But the paper then got too discursive and as we really want to highlight the binders in foods, we decided to leave the feed and farm stage parts completely out. We really want to focus on binders in humans, in foods, which has not been discussed previously at all. The dynamics, functionality and ethics regarding binders in human foods is completely different from aflatoxin control methods on farm levels, although all these are related to aflatoxins in the feed and food chain, the perspective and logic is very different. That is why we wanted to leave these feed discussions away from this paper and really focus only on binders and humans, and the problems related.

Lines 164 - 174: the description of a general method for aflatoxin binding studies should be placed in the beginning of a chapter not by the end.

Shifted to second column under the 3. chapter

Lines 198-199: Could you provide data on poor vs wealthy consumer have different exposure to aflatoxins through food? Do you have some numbers to provide a clear picture? What are the differences in aflatoxin contaminated foods between informal and formal markets? Are there data of the surveys for these two markets?

Data is limited, but we have some evidence especially in milk from Nairobi: (ref 23,54) showing how all milk is contaminated, on average milks in informal markets were found to have more aflatoxins than formal markets. And also, as formal markets are somehow controlled and monitored, but informal markets are not they are inherently difficult to monitor. This is what is being highlighted in lines 218-221.

What are the side effects of using aflatoxin binders in food and feed, especially for clay supplements and LAB?

This is tricky as there is very little done with these binders in humans, and some in animals (which we did not want to discuss as the focus is on humans); in the case of humans we argue the whole concept is unethical and questionable, so  it would be also questionable to find out the possible side effects.

Lines 233-240: Indicate what are the limits of aflatoxins in food in Europe, North America, and Africa.

Regulations are discussed 253-260 and references to regulations are provided – these vary depending on the region and commodity/food product, and is not the focus of the paper as binders are not related to the regulations directly (aflatoxins are still present in foods whether “bound” or not).

Reviewer 2 Report

Dear Author/s,

The paper deals with the highly urgent but difficult-to-solve topic: the aflatoxins in Africa. Although papers with this topic are often published, it deserves attention as the situation is still not getting better. The contamination of food and feed with aflatoxins in Africa is a serious problem from the human health point of view. There are many economic and social consequences. Specifically, paper is focused on possible detoxification strategies, mainly use of lactic acid bacteria (LAB) and discuss the positives and negatives of the introducing this strategy into real African (Kenyan) practice. The author/s regard using detoxification for agricultural products as counter-productive and don´t recommend it.

The big problem of the the paper is English.  Improper use of English sometimes obscure the meaning of the sentence or the whole paragraph. It is necessary to do carefull repeated proofreading. 

Line 19: Penicillium is not mycotoxin but fungi, mycotoxin producer. Please, correct.

Line 22: hepatitis instead of „hepatis“

The citations are not systematically replaced by numbers – in some parts of text there are authors and years instead of numbers (e.g. lines 65, 104, 107-108, check carefully the whole text, please).

Referee

Author Response

Dear Author/s,

The paper deals with the highly urgent but difficult-to-solve topic: the aflatoxins in Africa. Although papers with this topic are often published, it deserves attention as the situation is still not getting better. The contamination of food and feed with aflatoxins in Africa is a serious problem from the human health point of view. There are many economic and social consequences. Specifically, paper is focused on possible detoxification strategies, mainly use of lactic acid bacteria (LAB) and discuss the positives and negatives of the introducing this strategy into real African (Kenyan) practice. The author/s regard using detoxification for agricultural products as counter-productive and don´t recommend it.

The big problem of the the paper is English.  Improper use of English sometimes obscure the meaning of the sentence or the whole paragraph. It is necessary to do carefull repeated proofreading. 

English edit done to the manuscript, see the track changes version

Line 19: Penicillium is not mycotoxin but fungi, mycotoxin producer. Please, correct.

Changed to produced metabolite, patulin

Line 22: hepatitis instead of „hepatis“

Changed

The citations are not systematically replaced by numbers – in some parts of text there are authors and years instead of numbers (e.g. lines 65, 104, 107-108, check carefully the whole text, please).

Citations referring directly to these specific studies changed to format “Govaris [44]”

Humans have a right to food free from mycotoxins that could cause significant health risk – reference left as it is highlighting the direct quotation

Referee

Round  2

Reviewer 2 Report

I agree with changes, manuscript has been improved substantially.